# Perceptual learning of pitch provided by cochlear implant stimulation rate

**Susan R. S. Bissmeyer**[1,2]*, **Shaikat Hossain**[2], **Raymond L. Goldsworthy**[1,2]

**1** Department of Biomedical Engineering, Viterbi School of Engineering, University of Southern California, Los Angeles, California, United States of America, **2** Auditory Research Center, Caruso Department of Otolaryngology, Keck School of Medicine, University of Southern California, Los Angeles, California, United States of America

* ssubrahm@usc.edu

**Data Availability Statement:** All relevant data are within the manuscript and figures.

**Funding:** This work was supported by the USC Tina and Rick Caruso Department of Otolaryngology (RLG and SRSB), NIH NIDCD

## Abstract

Cochlear implant users hear pitch evoked by stimulation rate, but discrimination diminishes for rates above 300 Hz. This upper limit on rate pitch is surprising given the remarkable and specialized ability of the auditory nerve to respond synchronously to stimulation rates at least as high as 3 kHz and arguably as high as 10 kHz. Sensitivity to stimulation rate as a pitch cue varies widely across cochlear implant users and can be improved with training. The present study examines individual differences and perceptual learning of stimulation rate as a cue for pitch ranking. Adult cochlear implant users participated in electrode psychophysics that involved testing once per week for three weeks. Stimulation pulse rate discrimination was measured in bipolar and monopolar configurations for apical and basal electrodes. Base stimulation rates between 100 and 800 Hz were examined. Individual differences were quantified using psychophysically derived metrics of spatial tuning and temporal integration. This study examined distribution of measures across subjects, predictive power of psychophysically derived metrics of spatial tuning and temporal integration, and the effect of training on rate discrimination thresholds. Psychophysical metrics of spatial tuning and temporal integration were not predictive of stimulation rate discrimination, but discrimination thresholds improved at lower frequencies with training. Since most clinical devices do not use variable stimulation rates, it is unknown to what extent recipients may learn to use stimulation rate cues if provided in a clear and consistent manner.

## I. Introduction

In the auditory system, acoustic frequency is encoded in the place-of-excitation and timing properties of the auditory nerve response. Place coding of frequency is produced by cochlear mechanics and captured by auditory neurons giving rise to the well-established tonotopy of the auditory system [1]. Temporal coding of frequency is captured by phase-locked firing synchronous to acoustic frequencies at least as high as 3 kHz [2–6], and arguably as high as 10 kHz [7], although there is a considerable debate in the field on the exact limit of temporal frequency coding [8]. Cochlear implants use place and timing of stimulation by allocating higher

Grant T32 DC009975-10 (SRSB and SH) and NIH
NIDCD Grant R01 DC018701 (RLG and SRSB)
(https://keck.usc.edu/otolaryngology/; https://
dornsife.usc.edu/hcn/; https://www.nidcd.nih.gov/
;). The funders had no role in study design, data
collection and analysis, decision to publish, or
preparation of the manuscript.

**Competing interests:** The authors have declared
that no competing interests exist.

acoustic frequencies to more basal electrodes and by modulating constant-rate stimulation with temporal envelopes [9]. Historically, sound processing for cochlear implants has limited temporal cues to modulation frequencies less than 300 Hz [10]. Limiting stimulation timing in such a manner discards temporal fine structure [11], which if preserved might improve pitch perception and speech comprehension in noise for cochlear implant users [12–17]. The present study considers individual differences and perceptual learning for stimulation rate discrimination when provided in a clear and consistent manner using single-electrode stimulation.

Stimulation rate was one of the first psychophysical dimensions explored in cochlear implant science, and studies have shown that recipients hear increasing pitch with increasing stimulation rate, but resolution decreases above 300 Hz [18–25]. Several factors may contribute to the loss of resolution for higher rates, with lack of experience possibly a factor. In general, many aspects of hearing improve with experience. Speech comprehension and auditory awareness dramatically improve over the first year after cochlear implantation, and even after years of experience further benefits can be derived from auditory training [26–28]. Training can improve psychophysical abilities and speech comprehension [26,29,30]. Since variable stimulation rate is not typically used by clinical devices, it is possible that recipients could learn to use it if provided access to the new information. Efforts to restore temporal fine structure to cochlear implant stimulation have demonstrated mixed results, but certain studies indicate benefits for speech and music perception emerging with experience [9,12–14].

Perceptual learning has seldom been explored specifically for cochlear implant stimulation rate. Goldsworthy and Shannon (2014) found that rate discrimination improves with training, with benefits observed for rates as high as 3520 Hz [31]. This perceptual learning of stimulation rate as a cue for pitch is similar to that shown for normal hearing non-musicians, who can improve their frequency discrimination from 1 to 0.1%, equivalent to musician level performance, with training [32]. This musician advantage has been shown to be preserved across age and degrees of hearing loss [33]. Considering the remarkable plasticity of tonotopic coding of pitch [34], one might predict comparable plasticity of temporal pitch mechanisms [35].

Hypothetically, stimulation rate discrimination may also depend on the health of the auditory nerve. Rate discrimination varies across subjects and within subjects across electrodes [25,36,37]. Variations in neural health may limit stimulation rate sensitivity through diminished population coding and diminished neural synchrony. The present study considers individual differences in psychophysical measures of spatial tuning and temporal integration as predictors of stimulation rate discrimination. Spatial tuning is quantified through forward-masked detection thresholds and average detection thresholds, which reflect multiple aspects of spatial tuning including electrode-neural geometry, local neural health, and tonotopic pitch associated with different places of excitation [38–41]. Others have suggested that decoding of temporal fine structure in ascending auditory pathway depends on precise phase relationships across auditory nerve fibers, characteristics that diminish with impoverished neural health [42]. Degradation of the spatial patterning of neural health may degrade the temporally precise mechanisms observed in the cochlear nucleus that have been suggested as underlying encoding of temporal fine structure into average rate codes [42]. Consequently, forward-masked and detection thresholds are considered here to the test the hypothesis that degradations in spatial tuning will affect temporal pitch mechanisms that rely on precise integration times across fibers. Further, even in the absence of gross degradations of spatial tuning, it is hypothesized that temporal precision supported by fast-acting ion channels and vesicle release will contribute to individual differences observed in rate discrimination. Consequently, the psychophysically derived metric of multi-pulse integration is calculated from measured detection

thresholds (without masking) to quantify neural integration and potential neural degeneration [43,44].

There is some evidence that monopolar mode may improve performance for measures of intensity discrimination, speech recognition, and rate discrimination [38,45–47], while other studies have shown no consistent benefit from stimulation mode [48–51]. There has also been no agreed upon advantage for electrode location along the current electrode array, with a tendency toward an apical electrode location benefit [31,52–55], while other studies focus on finding local extrema at various individual locations along the array [38,40,43,44,56]. The configurations in the present study were focused on comparing bipolar to monopolar stimulation mode and the most distal electrode locations feasible along the array.

The present study was designed to examine individual differences and perceptual learning of stimulation rate discrimination when provided in a clear and consistent manner. Individual differences were examined to test the hypothesis that stimulation rate discrimination can be predicted by psychophysical measures of spatial tuning and temporal integration, which serve as a proxy for estimating the health of the auditory nerve. The effect of psychophysical experience over three test sessions was examined to test the hypothesis that rate pitch discrimination improves with focused psychophysical training. The results provide insight into the extent that variable stimulation rates could be used to improve cochlear implant pitch perception.

## II. Materials and methods

### A. Subjects

Seven cochlear implant users participated in this study. Four bilateral users were tested and completed the protocol in each ear sequentially, with the first ear tested randomly selected. All subjects were implanted with devices from Cochlear Corporation. Relevant subject information is provided in Table 1. Participants provided informed consent and were paid for their participation. The University of Southern California's Institute Review Board approved the study.

### B. Psychophysical testing

*Overview*. Subjects participated in a single-electrode psychophysical protocol with all procedures conducted using the USC Cochlear Implant Research Interface [57,58]. Procedures were scheduled during three sessions, with one week between sessions. All procedure used cathodic-leading biphasic pulse trains and always provided correct-answer feedback. Every measure was tested at two locations (apical, basal) for two stimulation configurations (bipolar, monopolar) with counterbalancing of all locations and configurations across test sessions. During the initial session, loudness growth functions, baseline rate discrimination thresholds, and forward-masked thresholds were measured. On the second session, training was provided for rate discrimination for two hours and rate discrimination and forward-masked thresholds were measured again. The third session was as the second session, with two hours of psychophysical training followed by rate discrimination and forward masking measures.

**i. Detection thresholds and comfort levels as a function of stimulation rate.** Detection thresholds and comfort levels were measured as a function of stimulation rate to provide loudness balancing in rate discrimination procedures. Detection thresholds and comfort levels were measured using a method of adjustment. Subjects used a graphical user interface with six sliders controlling different stimulation rates from 50 to 1600 Hz in octave intervals. After adjusting a slider, the subject would hear a 400 ms pulse train comprised of biphasic pulses with 50 μs phase durations and 50 μs interphase gaps, the stimulation rate corresponding to the slider and the current level corresponding to the slider height (values rounded to the

**Table 1. Subject demographics.**

| Subject | Gender | Ear tested | Etiology | Age at Onset of Hearing Loss (yrs) | Age at Deafness | Age at implantation | Age at time of testing | Bipolar Mode |
|---------|--------|-----------|----------|-----------------------------------|-----------------|---------------------|------------------------|--------------|
| C1 | M | Both | Meniere's | 39 | L:46 R:39 | L:46 R:43 | 46 | BP+3 |
| C2 | F | Both | Unknown | 15 | 22 | L:23 R:27 | 33 | BP+2 |
| C3 | F | Both | Progressive Nerve Loss | 40 | 53 | L:54 R:58 | 71 | BP+3 |
| C4 | M | Both | Progressive Nerve Loss | Birth | 7 | L:44 R:57 | 57 | BP+4 |
| C5 | M | Right | Noise Induced | 50 | 50 | 70 | 79 | BP+3 |
| C6 | F | Right | Progressive Nerve Loss | 20 | 50 | 64 | 67 | BP+3 |
| C7 | F | Left | Noise Induced | 20 | 44 | 54 | 66 | BP+1 |

nearest clinical unit). Subjects were instructed to adjust all six sliders to detection threshold and then to values that were loud but comfortable. The subjects were instructed to loudness balance their detection and loud but comfortable levels across frequencies. The bipolar mode chosen for each subject was the narrowest configuration in which they could still reach the loud but comfortable level (100% dynamic range). The resulting detection thresholds and comfort levels were fit with a logistic equation of the form:

$$Y(x) = U - \frac{U - L}{(1 + Qe^{-Bx})^{\frac{1}{v}}}, \tag{1}$$

where U and L are the upper and lower limits of the subject's dynamic range, Q is related to the current level at 100 Hz, B is the rate by which the current decreases over the frequency range, x is frequency expressed as $\log_2(\text{frequency}/100)$, and v controls asymptotic growth. Various equations were explored, and the fitted logistic equation provide the best compromise in terms of shape for the nonlinear decrease in levels for increasing rates and had the lowest adjusted mean squared error out of the functions considered. These were used to balance loudness in subsequent rate discrimination procedures.

**ii. Rate discrimination thresholds.** Rate discrimination thresholds were measured using a two-interval, two-alternative, forced- choice procedure in which subjects were asked to select the interval that was higher in pitch. Both the standard and target were 400 ms pulse trains comprised of biphasic pulses with 50 µs phase durations and 50 µs interphase gaps. The standard stimulation rates tested were nominally 100, 200, 400, and 800 Hz. The rate of the target stimulus was adaptively controlled. For each interval, the separate amplitudes of the standard and target were randomly roved between 90 and 100% (uniform distribution) of the subject's dynamic range as fitted by the logistic function.

The initial difference between standard and target stimulation rates was 40%. This difference was decreased by a step following correct responses and increased by three steps following incorrect responses (75% detection accuracy, Kaernbach, 1991) [59]. The initial step size was 0.9 and was decreased by a factor of $2^{-\frac{1}{2}}$ after each reversal until obtaining a value of $\sqrt{2}$ on the fourth reversal, at which point the step size was held constant at $\sqrt{2}$ (i.e., requiring two correct responses to halve the rate difference). Adaptive runs continued for a total of 8 reversals and the discrimination threshold was calculated as the average of the last 4 reversals.

**iii. Psychophysical training of stimulation rate discrimination.** Psychophysical training of stimulation rate discrimination was conducted using single-electrode rate discrimination procedures. The procedure used was a two-interval, two-alternative, forced-choice procedure

in which the stimulation rate difference between the standard and target intervals was held constant but the base rate was adaptively increased to provide training at increasing higher stimulation rates [31]. The standard and target stimuli were 400 ms pulse trains comprised of biphasic pulses with 50 μs phase durations and 50 μs interphase gaps. Stimulation current levels were controlled using the fitted logistic functions to detection thresholds and comfort levels. The initial value of the standard rate was 100 Hz and the target stimulation rate was specified to be 20% higher than the standard (i.e., 120 Hz). Stimulation rates were constrained between 100 and 1600 Hz. Following correct responses, both the standard and target stimulation rates were increased by a step; following incorrect responses, both the standard and target stimulation rates were decreased by three steps (75% detection accuracy) [59]. The initial step size was $2^{1/3}$ (i.e., the base rate doubled after 3 correct responses), but was reduced by a factor of 0.9 until obtaining $2^{1/12}$ (i.e., the base rate was increased by a semitone after each correct response). Adaptive runs continued for 12 reversals and the upper limit of discrimination was calculated as the average of the last 6 reversals. This procedure was conducted for 8 conditions consisting of the 4 combinations of apical/basal electrodes and monopolar/bipolar stimulation modes first tested using a 20% rate differences, then tested using a 10% rate difference. Total training time was approximately 2 hours.

**iv. Forward-masked detection thresholds.** Forward-masked detection thresholds were measured for combinations of masker locations (apical, basal) and stimulation configurations (bipolar, monopolar) using a three-interval, three-alternative, forced-choice procedure for a set of probe electrode locations. The masker and probe were 500 Hz pulse trains comprised of biphasic pulses with 50 μs phase durations and 50 μs interphase gaps. The masker and probe were 200 and 20 ms in duration, respectively. The probe was presented following the masker with the first phase of the probe starting 2 ms after the first phase of the last masker pulse. For this procedure, two of the intervals only contained the masker, while the randomly assigned target interval included the probe following the masker. The probe locations evaluated were 0, 1, 2 and 4 electrodes away from the masker electrode. The initial value of the probe stimulus was set to the subject's comfort level in clinical units. The level of the probe was decreased by a step following correct responses and increased by three steps following incorrect responses, which converges to 75% detection accuracy [59]. The initial step for a run was 8% of the subject's dynamic range in clinical units, and the step was decreased by a factor of $2^{-\frac{1}{2}}$ after each reversal until obtaining a value of 2% on the fourth reversal, at which point the step was held constant at 2% of the subject's dynamic range. An adaptive run continued for a total of 10 reversals and the forward-masked threshold for the run was calculated as the average of the last 6 reversals.

## C. Statistical methods

Stimulation rate discrimination thresholds were measured for all combinations of stimulation modes (bipolar, monopolar), electrode locations (apical, basal), and stimulation rates (100, 200, 400, 800 Hz). For each condition, rate discrimination thresholds were measured with three repetitions and test sessions were repeated once a week for three weeks. Stimulation rate training was administered each week through a staircase method which provided subjects training into higher rates depending on their ability to discriminate rates. The statistical method implemented for rate discrimination is a multi-factorial repeated-measures analysis of variance (ANOVA) with second-order interactions for the factors of subject, stimulation mode, electrode location, test session, and stimulation rate. All statistics were calculated on logarithmically transformed rate discrimination thresholds, with the rationale for using logarithmic transforms provided by Micheyl and colleagues [32]. Post-hoc multiple comparisons were implemented for significant factors and interactions.

Three psychophysically derived metrics of spatial and temporal tuning were calculated: detection thresholds, multi-pulse integration, and forward masking. All psychophysically derived metrics were calculated for each subject, stimulation mode (bipolar, monopolar), and electrode location (apical, basal). Normalized correlation coefficients were calculated for rate discrimination thresholds and for all psychophysically derived metrics. A coefficient of rate discrimination thresholds was calculated for each subject based on the average rate discrimination threshold across stimulation rates. A coefficient of detection thresholds was calculated as the mean value across stimulation rates. A coefficient of multi-pulse integration slopes was calculated based on linear regression of the measured detection thresholds for stimulation rates from 50 to 800 Hz. A coefficient of forward masking was generated through fitting forward-masked thresholds with a regression line to the probe locations separated by 0, 1, and 2 electrodes from the masker. All coefficients were normalized on a linear scale by subtracting the average across all conditions. Analysis of variance with second-order interactions was implemented for all metrics for the factors of subject, stimulation mode, and electrode location. Correlation analysis was implemented using a linear correlation procedure which analyzed the correlation coefficients and produced a R value and p value to characterize the linear orientation and confidence level of the correlation, respectively.

## III. Results

Results were collected for seven cochlear implant users, four of whom were bilateral and were tested in each ear. Each subject completed the three-session protocol including laboratory measures of rate discrimination training and assessment, and metrics such as forward-masked thresholds and equal-loudness contours as a function of stimulation rate. Results examine distribution of measures across subjects, the effect of psychophysical training on observed thresholds, and the predictive power of psychophysically derived metrics of spatial tuning and multi-pulse integration.

### A. Rate discrimination thresholds

Fig 1 shows rate discrimination thresholds for each subject averaged across repetitions and sessions. Rate discrimination thresholds vary greatly among subjects. Fig 2 shows individual and median rate discrimination thresholds to highlight trends across mode and electrode.

Analysis of variance was implemented on the measured rate discrimination thresholds with subject, stimulation mode, electrode location, stimulation rate, and test session (i.e., week) as factors. Subject was significant indicating substantial variability across subjects in terms of average discrimination thresholds ($F_{(10, 1479)} = 13.23$, $p < 0.001$). Stimulation mode was not significant indicating that average rate discrimination thresholds were not statistically different between bipolar and monopolar stimulation modes ($F_{(1, 1479)} = 0.2$, $p = 0.6827$). Similarly, electrode location was not significant indicating that average rate discrimination thresholds were not statistically different between apical and basal electrode locations ($F_{(1, 1479)} = 0.93$, $p = 0.3359$). Stimulation rate, as expected given the noted deterioration of discrimination with increasing stimulation rate, was significant ($F_{(3, 1479)} = 589.6$, $p < 0.001$). Test session was significant ($F_{(2, 1479)} = 6.16$, $p = 0.0022$) indicating that average rate discrimination thresholds improved over the course of the three-session protocol.

All second order interactions with subject were highly significant ($p < 0.001$) indicating substantial individual variability in how thresholds were affected by stimulation mode, electrode location, and auditory training. The interaction between stimulation mode and electrode location was weakly significant ($F_{(1, 1479)} = 4.49$, $p = 0.0343$). Post-hoc multiple comparison indicated that this interaction between stimulation mode and electrode location was primarily

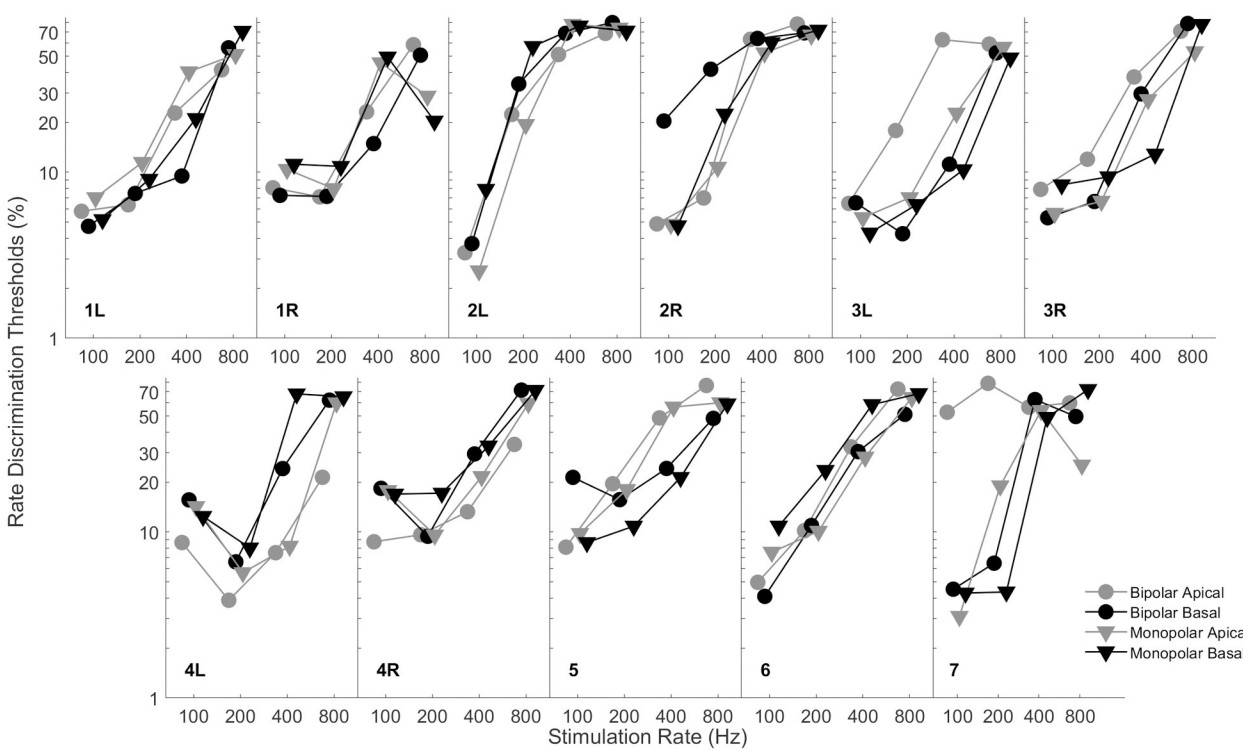

**Fig 1. Individual rate discrimination thresholds by mode and electrode across frequencies.**

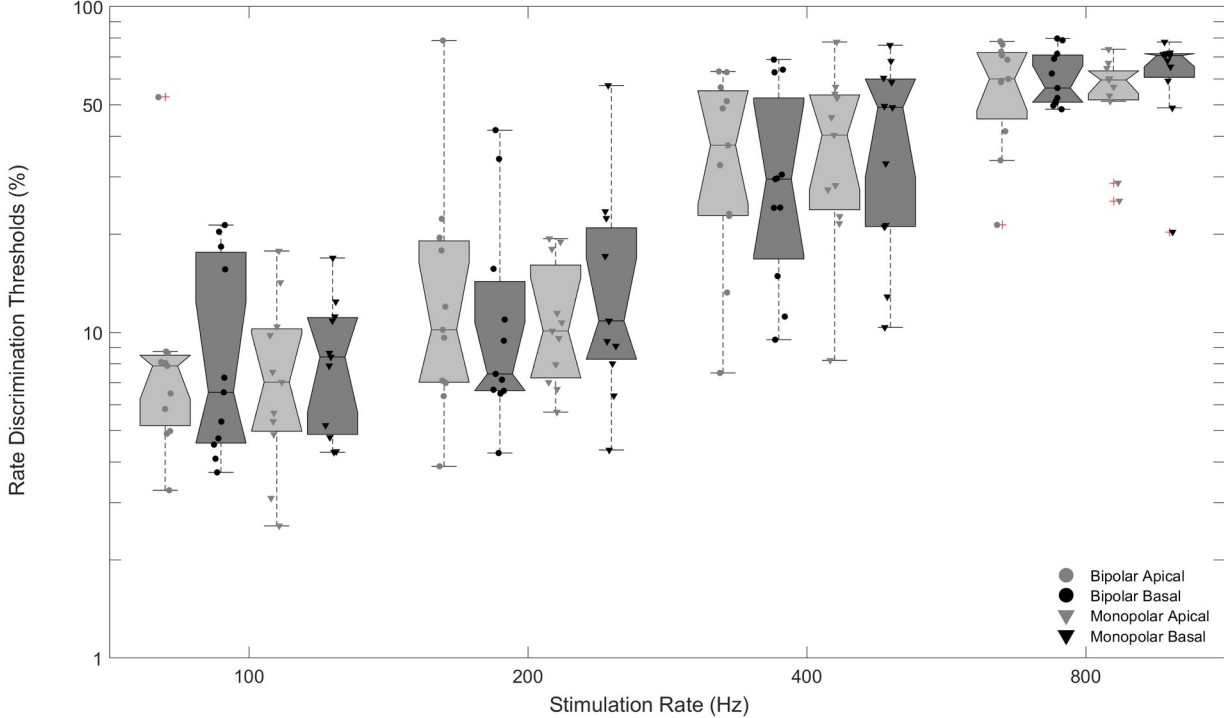

**Fig 2. Boxplot showing median rate discrimination thresholds by mode and electrode across frequencies.** Filled symbols represent each implant user's thresholds over weekly session and repetition.

driven by lower discrimination thresholds observed for monopolar stimulation of the apical electrode. Generally, there has been mixed discussion in the literature on an established stimulation mode or electrode location benefit, with a tendency toward monopolar mode and apical location stimulation providing a benefit, which could drive the interaction between monopolar stimulation and apical electrode location [38,49,52].

The highly significant interaction between both subject and mode and subject and electrode location revealed that although there was not a group benefit for stimulation mode or electrode location, some individuals were affected by mode and location. Post-hoc multiple comparisons revealed that although some subjects received a significant benefit for a specific mode or electrode location as gauged by Fisher's least-significant difference criteria, the majority of these significant results were not significant as gauged by more stringent Bonferroni criteria.

## B. Rate discrimination improves through experience

Rate discrimination thresholds improved significantly over the course of the three-session protocol ($F_{(2, 1479)} = 6.16$, $p = 0.0022$). Fig 3 compares the results of the present study to those of Goldsworthy and Shannon (2014) [31]. Rate discrimination thresholds are plotted versus hours of psychophysical training for different base rates. The results of the present study showed that training over three sessions with one week between each session provides a significant performance benefit to implant users. Discrimination thresholds significantly improved from week 1 to week 3 with averaged thresholds being 21.5% before training and 18.26% after training, improving on average by 1.62% per week.

## C. Forward-masked detection thresholds

Forward-masked thresholds were measured for specific probe locations separated by 0, 1, 2, and 4 electrodes from the masker electrode. Fig 4 shows forward-masked thresholds for the forced-choice procedure for apical and basal electrode locations and for bipolar and monopolar stimulation modes. Thresholds were normalized in linear scale by dividing the forward masking function by the peak threshold shift resulting in scale where 0 is no masking and 1 is maximum masking [39]. Thresholds typically decreased monotonically with increasing separation between masker and probe electrodes, but with noted deviations from that rule as observed for subjects 2L, 2R, 4L, and 4R.

**i. Psychophysically derived metric: Forward-masked threshold slopes.**   Forward-masked thresholds generally saturated at a spatial separation of 2 electrodes, so the furthest electrode was excluded from the fitted slopes. Subject was significant ($F_{(10, 10)} = 9.92$, $p < 0.001$) indicating individual differences in spatial selectivity. Stimulation mode ($F_{(1, 10)} = 3.11$, $p = 0.1083$) and electrode location ($F_{(1, 10)} = 2.78$, $p = 0.1264$) were not significant factors. Second order interactions between subject and stimulation mode ($F_{(10, 10)} = 1.97$, $p = 0.1503$) and subject and electrode location ($F_{(10, 10)} = 1.86$, $p = 0.1719$) were not significant. The interaction between stimulation mode and electrode location ($F_{(1, 10)} = 12.21$, $p = 0.0058$) was significant. A post-hoc multiple comparison test indicated that this interaction between stimulation mode and electrode location was primarily driven by steeper forward-masked threshold slopes observed for bipolar stimulation of the apical electrode.

## D. Detection thresholds and comfort levels as a function of stimulation rate

Fig 5 shows detection threshold levels as a function of stimulation rate. Detection thresholds and comfort levels typically exhibit temporal integration across pulses by decreasing monotonically for increasing rates.

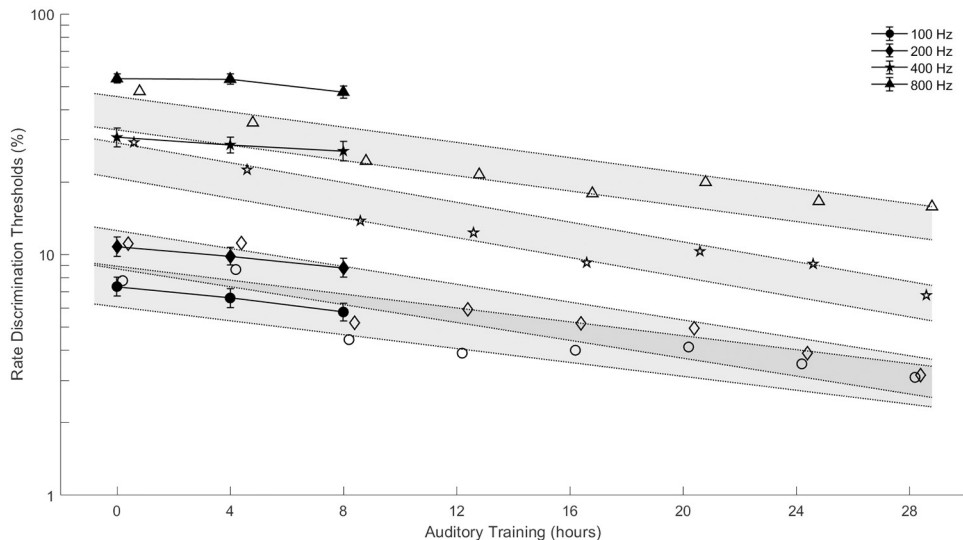

**Fig 3. The effect of training on rate sensitivity in current study, closed symbols, compared to rate training data in Goldsworthy and Shannon (2014), open symbols, with a linear fit to the data shown with the root-mean-square error in shaded gray.**

**ii. Psychophysically derived metric: Average detection thresholds.** Average detection thresholds were calculated from the detection thresholds shown in Fig 5. Group averages indicate similar thresholds for apical and basal electrodes when tested in monopolar mode, but with apical sites having higher average thresholds than basal sites when tested in bipolar mode. Subject was significant indicating the variability across subjects in stimulation levels required

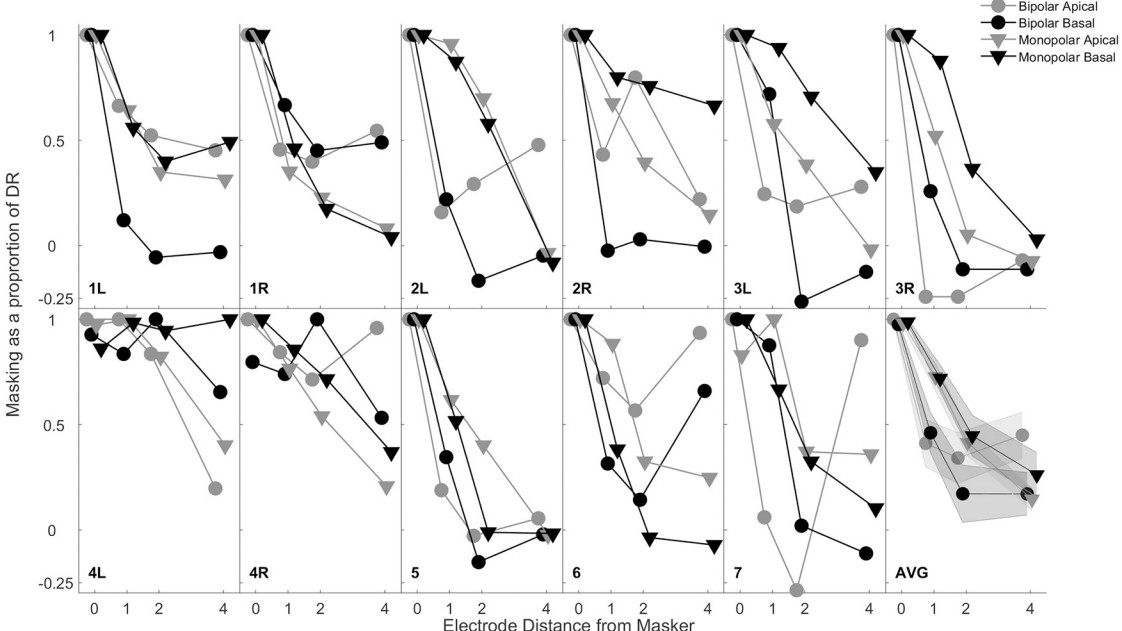

**Fig 4. Forward masking protocol with individual results for CI users.** These thresholds are represented as a proportion of percent dynamic range relative to the peak of the masking function.

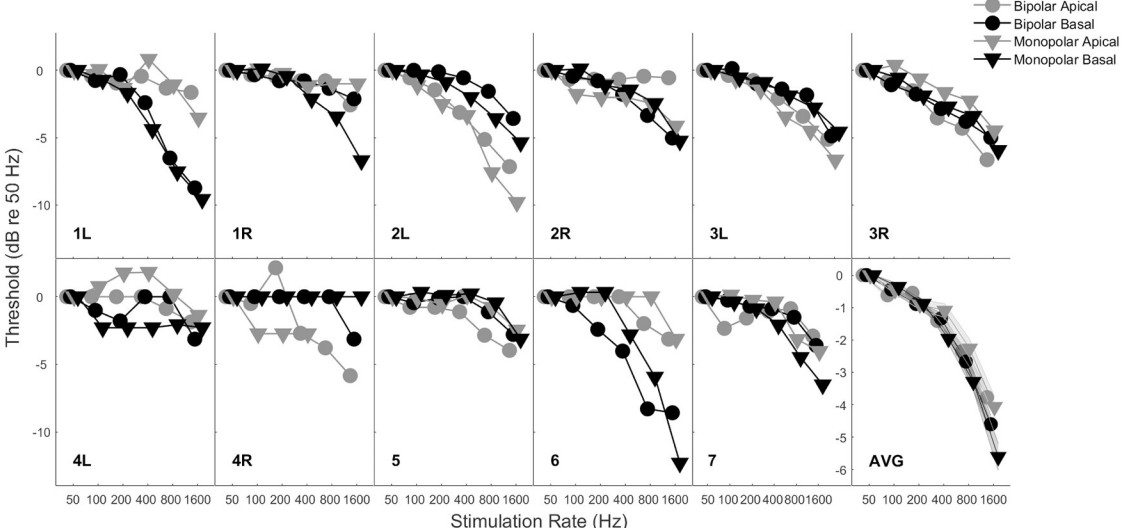

**Fig 5. Individual threshold levels across stimulation rate exhibiting integration across pulses.**

to obtain detection thresholds ($F_{(10, 10)}$ = 9.6, p < 0.001). Stimulation mode was significant reflecting the lower stimulation levels required to reach detection thresholds for monopolar stimulation ($F_{(1, 10)}$ = 34.9, p < 0.001). Electrode location was not significant ($F_{(1, 10)}$ = 2.0, p = 0.18) indicating similar thresholds for apical and basal stimulation sites. The interaction between subject and stimulation mode was not significant ($F_{(10, 10)}$ = 1.8, p = 0.17); however, the interaction between subject and electrode location was significant indicating individual variability in terms of apical and basal stimulation levels required to reach threshold ($F_{(10, 10)}$ = 7.9, p = 0.0015). The interaction between stimulation mode and electrode location was not statistically significant ($F_{(1, 10)}$ = 3.2, p = 0.10).

**iii. Psychophysically derived metric: Multi-pulse integration.** Multi-pulse integration is shown in Fig 5 as a function of the decrease in measured detection thresholds due to integration across pulses with increasing stimulation rate. In contrast to the other metrics, subject was not significant ($F_{(10, 10)}$ = 2.2, p = 0.11) indicating lower variability in subject performance across conditions than was the case for either average thresholds or forward-masked tuning slopes. Neither stimulation mode ($F_{(1, 10)}$ = 2.6, p = 0.14) nor electrode location ($F_{(1, 10)}$ = 0.1, p = 0.76) were significant. Neither the interaction between subject and stimulation mode ($F_{(10, 10)}$ = 1.1, p = 0.44) nor between stimulation mode and electrode location ($F_{(1, 10)}$ = 0.04, p = 0.84) were significant. The interaction between subject and electrode location, however, was significant ($F_{(10, 10)}$ = 5.2, p = 0.008) indicating that individual differences in multi-pulse integration slopes across apical and basal stimulation sites may be promising as a potential predictor of individual variability.

## E. Correlation analysis between psychophysically derived metrics and rate discrimination

The three psychophysically derived metrics (forward-masked threshold slopes, detection thresholds, and multi-pulse integration) were analyzed for correlation with rate discrimination thresholds. These metrics were calculated for each subject for each combination of stimulation mode, electrode location, and stimulation rates. We first tested the correlation between grand averages across stimulation modes, electrode locations, and stimulation rates for each metric

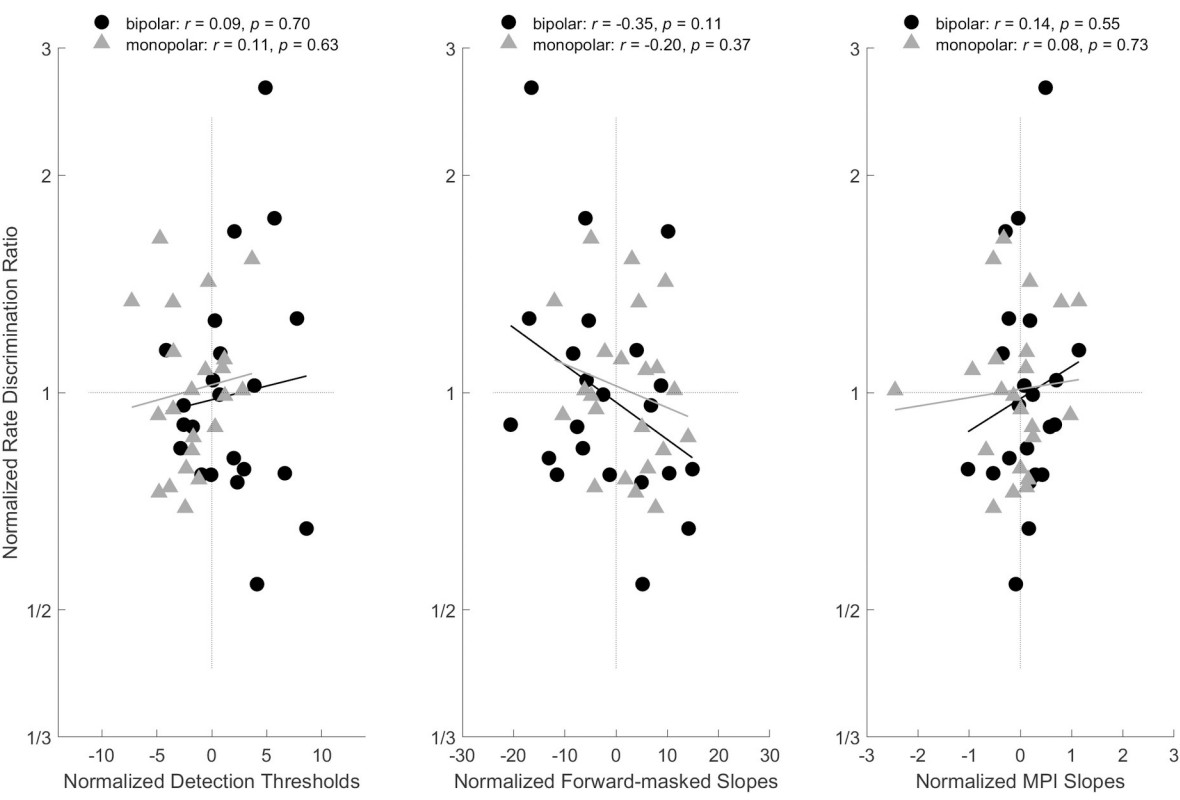

**Fig 6. Correlation analyses for rate discrimination correlated with detection thresholds, forward-masked slopes, and multi-pulse integration slopes normalized to the average across conditions for each subject.**

and the corresponding grand average measured rate discrimination. None of those correlations were significant. As suggested by Zhou and Pfingst (2016a, 2016b), such across-subject correlations tend to be weak since multiple individual differences affect perceptual outcomes [43,44]. Consequently, we then tested the correlation between normalized metric differences and normalized rate discrimination differences. Specifically, to normalize the average across conditions of the metrics and rate discrimination thresholds for each subject were subtracted from the metrics and discrimination thresholds for each subject for each electrode and stimulation mode (Fig 6). None of the correlations tested were significant at a 0.05 level, let alone the more stringent levels recommended for multiple comparisons analyses. In the correlation between forward-masked threshold slopes and rate discrimination thresholds, the apical electrode yielded a weak negative correlation ($p = 0.06$) indicating slightly lower rate discrimination thresholds for shallower forward-masked slopes, in agreement with literature correlating shallower forward-masked slopes with neural health [43,44]. Consistent with the literature, there is a slight trend toward steeper multi-pulse integration slopes correlating with lower rate discrimination thresholds [38].

## F. Exploratory correlation analysis among psychophysically derived metrics

Exploratory correlation analyses performed among the psychophysically derived metrics (forward-masked threshold slopes, detection thresholds, and multi-pulse integration) to analyze within-category correlations yielded mildly significant results. The correlations were done by

normalizing by the average across conditions for each subject. In the correlation between forward-masked threshold slopes and detection thresholds, monopolar mode yielded a weak positive correlation ($p = 0.132$) and basal electrode yielded a significant positive correlation ($p = 0.003$) indicating slightly lower detection thresholds for steeper forward-masked threshold slopes, consistent with the literature [40,44]. The correlation between multi-pulse integration slopes and detection threshold produced a significant result for both bipolar ($p = 0.016$) and monopolar ($p = 0.003$) stimulation modes and for the basal electrode ($p = 0.015$) with higher detection thresholds correlating with steeper multi-pulse integration slopes. Forward-masked threshold slopes exhibited a weak correlation to multi-pulse integration slopes for bipolar mode ($p = 0.08$) and a stronger correlation for the basal electrode ($p = 0.01$) with shallower forward-masked threshold slopes correlating with steeper multi-pulse integration slopes, consistent with the literature [43,44].

## IV. Discussion

The present study was designed to examine individual differences and perceptual learning of stimulation rate discrimination in adult cochlear implant users. Individual differences were examined by measuring rate discrimination at different electrode sites and configurations, while exploring psychophysically derived metrics of spatial tuning and temporal integration. While much across and within subject variability was observed for rate discrimination and for derived metrics, few significant correlations were observed. One observed trend, consistent with previous studies [38], was that stimulation rate discrimination tended to be better for monopolar stimulation in the apex. Perceptual learning of rate discrimination was examined across several training sessions and discrimination thresholds improved with training. That stimulation rate discrimination improves with experience suggests that the true limits of rate discrimination may remain unknown until clinical devices provide such information in a clear and consistent manner.

### A. Comparison with stimulation rate discrimination literature

For the lower stimulation rates tested (100, 200 Hz), the discrimination thresholds of the present study are comparable to the more sensitive thresholds reported in the literature. In the present study, the average discrimination threshold prior to training was 8% for 100 Hz base rates. The more sensitive thresholds reported in the literature are typically between 7 and 10% at 100 Hz [24,31,60,61]. The review by Moore and Carlyon (2005) was a compilation of 5 studies across 19 subjects [20,25,60,62–64]. Other studies found poorer discrimination, with thresholds measured at 100 Hz ranging from 20 to 40% [65–67].

For the higher stimulation rates (400, 800 Hz), discrimination thresholds of the present study are more sensitive than generally reported. Average rate discrimination thresholds of the present study were 30% and 54%, when measured at 400 and 800 Hz, respectively. In comparison, Townshend and colleagues (1987) reported that two of their subjects had an average 40% rate discrimination threshold, while the third subject could not discriminate rates above 175 Hz [24]. Three studies reported rate discrimination thresholds in the range of 47–49% [25,61,66]. Bahmer and Baumann (2013) reported thresholds of 69.2% [65]. Few studies tested single-electrode rate discrimination above 400 Hz. Bahmer and Baumann (2013) reported 76.9% at 566 Hz and Zeng (2002) reported 113% for 500 Hz [25,65]. Two of the subjects in Townshend and colleagues (1987) could discriminate rates above 400 Hz [24]. One of their subjects was able to discriminate 19% differences at 900 Hz and the other 15% differences at 1000 Hz. McKay and colleagues (2000) tested at 500 and 600 Hz, but their subjects could not discriminate rates based on perceived pitch [20].

The duration and amount of feedback seems to be a primary factor driving differences in measured rate discrimination thresholds across studies. Thresholds measured in the current study are consistent with most studies at lower frequencies, but more sensitive than studies which use a brief task exposure time. The current study is based on a total of 12 hours of psychophysical training and assessment with feedback always provided. Training was designed to gradually increase exposure to higher stimulation rates. The extended and progressive experience provides familiarity with the rate cue that is reflected in lower discrimination thresholds (Figs 1–3). Differences between the training protocol used in the present study and the training protocol used in Goldsworthy and Shannon, 2014, led to different amounts of training for the highest rates under consideration. Specifically, in the prior study from our group, training was provided also using an adaptive procedure to gradually introduce higher stimulation rates to subjects; however, the protocol used in Goldsworthy and Shannon, 2014, allowed for 32 reversals during each training run, which allowed subjects to work into progressively higher rates and hold their performance in that region. For the present study, only 12 reversals were provided during training, consequently, subjects received considerably less time to work and sustain into the higher rate region. This is the likely cause for why benefits of training were only observed for the lower rates in the present study.

## B. Lack of correlation between rate discrimination and other psychophysical measures

In the present study, psychophysically derived metrics of spatial tuning and temporal integration were not predictive of rate discrimination thresholds, but were consistent with previous studies [38,40,43,44]. We interpret these results as evidence that stimulation rate, at least for rates as high as 400 to 800 Hz, are well encoded into auditory nerve activity. It is difficult to know how much of the correlations are driven by the plasticity of pitch and the underlying peripheral sensitivity. It is quite possible that over the course of training, better correlations could be found as we explore the peripheral limitations, however, in the present study, pre- and post-training rate discrimination thresholds as well as thresholds at different base rates were examined for correlations and the conclusions remained the same. Studies using neural response telemetry have provided evidence that temporal synchrony of neural response is well maintained in cochlear implant recipients at least up to 1 kHz [3,31,68,69]. If forward-masked thresholds or multi-pulse integration quantify neural health, typical variations do not appear to strongly affect stimulation rate discrimination. A study which made a similar comparison found a weak relationship between multi-pulse integration slopes and rate discrimination [38]. We interpret the lack of correlation in a positive manner, that modest variations in neural health do not significantly impair a recipient's ability to hear pitch evoked by stimulation rate.

In the present study, the psychophysically derived metrics of spatial tuning and multi-pulse integration were calculated using the most apical and most basal electrode locations. Other studies examining metrics of spatial tuning and temporal integration of have explicitly chosen electrodes with steep and shallow slopes to consider effects at local extrema [43,44]. That approach may be more sensitive for detecting correlations since it may identify particularly healthy or damaged regions of the auditory nerve. Similarly, other variations in the measurement of forward-masked thresholds may affect the overall strength of correlation [39,44,57,70,71].

A significant interaction between stimulation mode and electrode location occurred for three of the measures: rate discrimination, forward-masked thresholds, and detection thresholds. One significant point was that the lowest rate discrimination thresholds and the shallowest forward-masked threshold slopes occurred for the same electrode-mode combinations of

bipolar basal and monopolar apical. This trend of lower rate discrimination thresholds correlating to shallower forward-masked threshold slopes, and thus steeper multi-pulse integration, agrees with the literature [38,44]. For detections thresholds, monopolar threshold levels were similar across electrode locations, and unsurprisingly were lower than bipolar thresholds. Detection thresholds in the bipolar configuration were lower in the base correlating with shallower forward-masked threshold slopes and lower rate discrimination thresholds for the same configuration and location. Overall, this provides a trend with monopolar providing better performance in the apex and bipolar providing better performance in the base. This trend is in agreement with Zhou and colleagues (2019) with the sites exhibiting the shallowest forward-masked thresholds slopes, monopolar apical and bipolar basal, providing improved rate discrimination performance over the sites with steeper forward-masked threshold slopes [38]. As mentioned, in the literature there has been a tendency toward a benefit provided by monopolar mode and apical location stimulation providing a benefit, which may contribute to the better performance in the monopolar apical configuration [38,49,52].

A limitation of the present study concerns how electrode configuration might affect rate discrimination in that the comparison was made between monopolar with relatively broad bipolar configurations. We chose to examine bipolar configurations for which comfort levels could be mapped with relatively short pulsatile phase durations. We chose to do so to concentrate charge in a temporally precise manner but doing so required broader bipolar configurations to be used. Consequently, it is unclear whether the effect of electrode configuration would be more pronounced using narrower configurations such as tripolar or quadrupolar. Another possible limitation is some subjects exhibited a non-monotonic pattern in their detection thresholds which can be observed in Fig 5. This non-monotonic pattern was reflected in the multi-pulse integration metric as well and was most pronounced for subject 4. The subjects were instructed to set their threshold as the lowest level at which they first heard the stimulus, with careful attention given to loudness balancing. That being the case, the detection thresholds may have been more conclusively set by another method, such as an alternative forced choice procedure.

## C. Psychophysical training improves stimulation rate discrimination

Stimulation rate discrimination improved with training. While pitch discrimination has been shown to be perceptually plastic even in normal-hearing listeners, it is possible that there is greater potential for training pitch associated with stimulation rate since variable stimulation rates are typically not used by cochlear implants. Few studies have considered perceptual learning of stimulation rate, though several studies have consider perceptual and physiological plasticity associated with tonotopy, with attention given to the tonotopic mismatch between the acoustically and electrically stimulated areas. Reiss and colleagues (2014) showed plasticity in the representation of place pitch provided by the frequency allocation of the cochlear implant processor, especially over the first two years of use [34].

Animal studies have considered temporal coding of frequency following deafening and implantation. Fallon and colleagues (2014) studied cats who were deafened at birth, implanted at 8 weeks, and activated 2 weeks post-surgery, after which they were stimulated constantly for 6–8 months [72]. They found that a moderate duration of deafness with cochlear implantation had minimal overall effect on the temporal response properties of neurons, with the only significant effect being the decreased ability of the neural population to respond to every pulse in a pulse train. Another study showed that longer durations of deafness can have more adverse effects on temporal response properties, but that training can provide a profound improvement in degraded temporal processing [73].

Given the evidence for plasticity of stimulation rate pitch perception and the evidence for strong neural synchrony to electrical stimulation, we speculate that variable stimulation rates can be used to improve pitch perception for cochlear implant users [31,74]. Attempts to restore temporal fine structure into cochlear implant stimulation have been mixed with some studies indicating no benefits, but others indicating benefits for speech and music perception that emerge over a year or more of experience [9,12,75,76]. A challenge associated with cochlear implant sound processing design is to determine the extent that temporal coding is limited by sound processing rather than by physiology. In that regard, single-electrode psychophysics are insightful as to the physiological limits and the potential for perceptual learning. Psychophysical training of stimulation rate pitch perception has rarely been investigated since measures require laboratory hardware and repeat visits from subjects. Typically, rate discrimination is assessed in acute laboratory protocols across one or two sessions without substantial familiarization or dedicated psychophysical training [18,37,77]. Since cochlear implant signal processing may not adequately use stimulation rate to encode acoustic cues, the only experience cochlear implant users may have with this cue for higher rates is during these acute laboratory visits designed to assess its salience.

For learning to occur, in general, the cue of interest must be presented in a clear and consistent manner and provided on a regular basis [78,79]. The results of the present study indicate that rate discrimination improves with as little as 12 hours of training and assessment. These results are consistent with Goldsworthy and Shannon (2014), which examined the effects of auditory training on rate discrimination thresholds of six cochlear implant users over the course of 28 hours of training and assessment, but with notably less improvement for the higher rates tested [31]. While the training in Goldsworthy and Shannon (2014) focused on rates from 110 to 1760 Hz, the training in the present study focused on the regions below 400 Hz due to the smaller rate differences used (10 & 20%), and interestingly the effects of training did not transfer to higher frequencies. Psychophysical training with large rate differences (e.g., > 40%) and using large number of trials (e.g., > 40) in the adaptive procedures would allow subjects to gradually work up to higher stimulation rates for consistent training at those rates. A consistent, daily training of the relevant rate cues under the right conditions may produce results similar to Goldsworthy and Shannon (2014).

## V. Conclusions

The present study examined individual differences and perceptual learning of stimulation rate pitch perception. Individual differences between and within subjects based on forward-masked thresholds and multi-pulse integration were not predictive of rate discrimination thresholds. We interpret this finding as evidence that peripheral coding of stimulation rate does not strongly affect rate discrimination in cochlear implant users. In contrast, rate discrimination thresholds significantly improved for base rates of 100 and 200 Hz with relatively brief exposure and training for associating pitch with stimulation rate. This provides further evidence for the plasticity of temporal pitch provided by stimulation rate. Consequently, sound processing strategies designed to encode acoustic temporal fine structure into fine timing of stimulation should be examined in the context of perceptual learning of pitch.

## Acknowledgments

The authors would like to thank our cochlear implant subjects who worked tirelessly on testing and training. The authors express our thanks to Melissa Wilson (PhD, MPH) for reviewing the manuscript and significantly contributing to our statistical analyses. The authors thank both reviewers and Dr. Ifat Yasin for providing detailed comments which greatly improved the

manuscript. Portions of this article were presented at the 2017 Conference on Implantable Auditory Prostheses, Lake Tahoe, California, July 20, 2017 and at the 175th Meeting of the Acoustical Society of America, Minneapolis, Minnesota, May 10, 2018.

## Author Contributions

**Conceptualization:** Susan R. S. Bissmeyer, Shaikat Hossain, Raymond L. Goldsworthy.

**Data curation:** Susan R. S. Bissmeyer.

**Formal analysis:** Susan R. S. Bissmeyer, Shaikat Hossain, Raymond L. Goldsworthy.

**Funding acquisition:** Raymond L. Goldsworthy.

**Investigation:** Susan R. S. Bissmeyer, Raymond L. Goldsworthy.

**Methodology:** Susan R. S. Bissmeyer, Raymond L. Goldsworthy.

**Project administration:** Susan R. S. Bissmeyer, Raymond L. Goldsworthy.

**Resources:** Susan R. S. Bissmeyer.

**Supervision:** Shaikat Hossain, Raymond L. Goldsworthy.

**Validation:** Susan R. S. Bissmeyer.

**Visualization:** Raymond L. Goldsworthy.

**Writing – original draft:** Susan R. S. Bissmeyer.

**Writing – review & editing:** Susan R. S. Bissmeyer, Raymond L. Goldsworthy.

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
