## [Decision Letter · Decision Letter 0]

18 Dec 2019

PONE-D-19-23873

Perceptual Learning of Pitch Provided by Cochlear Implant Stimulation Rate

PLOS ONE

Dear  Dr. Bissmeyer

Thank you for submitting your manuscript to PLOS ONE. After careful consideration, we feel that it has merit but does not fully meet PLOS ONE’s publication criteria as it currently stands. Therefore, we invite you to submit a revised version of the manuscript that addresses the points raised during the review process. Both reviewers suggested a major revision, and advise in particular, clarifying your statistical anlayses and providing more detail in the description of your results.

We would appreciate receiving your revised manuscript within 45 days of the date of this decision. To enhance the reproducibility of your results, we recommend that if applicable you deposit your laboratory protocols in protocols.io, where a protocol can be assigned its own identifier (DOI) such that it can be cited independently in the future. For instructions see: http://journals.plos.org/plosone/s/submission-guidelines#loc-laboratory-protocols

We look forward to receiving your revised manuscript.

Kind regards,

Ifat Yasin

Academic Editor

PLOS ONE

Journal Requirements:

http://www.journals.plos.org/plosone/s/file?id=wjVg/PLOSOne_formatting_sample_main_body.pdf and http://www.journals.plos.org/plosone/s/file?id=ba62/PLOSOne_formatting_sample_title_authors_affiliations.pd

Reviewers' comments:

Reviewer's Responses to Questions

**Comments to the Author**

1. Is the manuscript technically sound, and do the data support the conclusions?

Reviewer #1: Yes

Reviewer #2: Partly

2. Has the statistical analysis been performed appropriately and rigorously? 

Reviewer #1: No

Reviewer #2: I Don't Know

3. Have the authors made all data underlying the findings in their manuscript fully available?

Reviewer #1: Yes

Reviewer #2: Yes

4. Is the manuscript presented in an intelligible fashion and written in standard English?

Reviewer #1: Yes

Reviewer #2: Yes

5. Review Comments to the Author

Reviewer #1: The study examined individual variability in rate discrimination in cochlear implant users, and reported whether the variability can be attributed by spatial selectivity of neural excitation as well as the MPI functions. The paper also looked at the effect of training on rate discrimination. There were a lot of data, however, I found some of the results difficult to interpret because of several potential methodological issues. Introduction and methods can be improved by providing more in-depth rationale for the psychophysical measures chosen to be analyzed, choice of parameters, and the normalization procedures. Please find the more detailed comments below.

Re Introduction:The rational of relating spatial tuning to rate discrimination needs to be elaborated. There is a brief discussion of this later in the Method section, but it should be moved here. Why do the authors think spatial tuning should be related to rate discrimination? Is it based on the assumption that sharper spatial tuning should indicate higher neural density? We know however, electrode position is also a strong predictor.  

The authors used a general term, temporal tuning, to describe the effect of stimulation rate on thresholds. I think the term might be too broad. The mechanism undying the effect of rate on threshold might be completely different from that undying other temporal acuities, for example modulation sensitivity.

Methods:

Maybe I missed it but I don't see where the results of the masked thresholds using method of adjustment were reported. I wasn't quite sure that I understand why the masked thresholds were measured with method of adjustment at all. 

I don't quite agree with the way that the masking patterns were quantified. First, I think that the amount of masking, rather than the absolute masking thresholds should be used. That is, the masked thresholds should be considered in terms of how much they have elevated from the unmasked values. You might have a very high masked threshold, but that does not necessarily mean that the maker has exerted a great amount of masking on this probe. Second, I don't quite follow how the masking functions were normalized. It appears that the masked threshold in % DR at the 0 position was subtracted from all data points. Based on methodology paper from MaKay, the masking function (amount of masking against probe masker separation) should be divided by the peak of the function, rather than that at the 0 position. If the authors disagree with these methods, then I think the rationale should be provided.

It seems that each subject used a different electrode spacing for BP stimulation. I think it would be helpful to justify using different test parameters. I also wonder if this could have made it difficult to find an effect of stimulation mode?

Results:Figure2: it is unconventional to include results or stats in figure captions. Y label is not immediately clear and should be explained in methods.

The introduction section did not introduce the average detection thresholds as a part of the psychophysically derived metic. What was the rationale? I suspect that thresholds averaged across so many conditions, rates, location, and modes, would offer little information about the stimulation pattern or neural condition of the subjects. For example, we know thresholds at low versus high rates, and thresholds in MP versus BP stimulation would reveal completely different information about a stimulation site.

For some subjects and conditions, non-monotonic patterns were observed for the integration functions. These are unexpected, which makes me wonder, if the method of adjustment was accurate enough or estimating thresholds?

Line 265-267: if the effect of subject was not significant, wouldn't it indicate less subject variability?

Line 299-302: very interesting finding and I think it warrants some discussion

Line 311-316: This paragraph is not clear. Considering rewording. Did the authors mean "as shown in Fig 3"?  What is a significant result at the individual level?

Figure 4: figure caption doesn't seem right. 

The within-subject correlations may be done a little differently. For each measure, the data can be normalized to the average across conditions for each subject, thus removing between-subject variations. Correlations using differences between the apical and basal electrode certainly also removed the subject factor, but the within-subject effect of electrode condition was also removed. Also, maybe the authors would find the hypothesized relationships using thresholds at the higher base rates.

Line 352-360: These results were consistent with previous studies, and they indicate that the masked excitation patterns (related to spatial tuning) is negatively correlated with integration, so sharp spatial tuning and temporal tuning can't both indicate good neural condition. These need to be discussed in Introduction with more clearly laid out hypotheses.

 line 366-374: Do conclusions change if the post-training thresholds were considered for the various effects and correlations examined earlier?

A general comment: some of the degree of freedom reported do not look right to me. Please check.

Discussion:Line 430-432: please see my comments above regarding the relationship between spatial and temporal tuning - they are negatively correlated.

The current results showed that learning did not happen for higher base rates, but results from the earlier study from the same group showed that discrimination of extremely high rates can be learnt. This discrepancy should be discussed.

Reviewer #2: This paper describes an experiment to better understand pitch perception in CI users based on rate of stimulation. Pitch perception was measured over several sessions, providing participants with practice using a cue that is not provided by their everyday CI processor. Other psychoacoustic measures – forward masking and multi-pulse integration -- were carried out to evaluate whether neural health was correlated with the potential for improved rate discrimination with practice. Data are consistent with the conclusion that there are practice effects for rate pitch discrimination, at least for low standard rates. Little or no learning occurred for 400 and 800 Hz (in contrast to the previous dataset), and no predictors of individual differences were found.

The topic of this research – rate discrimination abilities of CI users – is interesting and could have clinical implications. The experiment is sensible and appears to have been executed appropriately. The fact that no predictors were found does not diminish by interest in these data, as I think we have learned something by looking for such an association. The paper is well written at the level of the sentence and paragraph. My comments focus primarily on the overall organization, statistics, and interpretation of results.

Although the introduction lays out a clear argument for the research design, with hypotheses that are clearly focused on rate pitch discrimination, the results section does not maintain that organization. The results begin with the secondary measures, and rate discrimination only appears half-way through this section. There are also extensive statistical analyses of the secondary measure that do not appear to be directly motivated by the hypotheses. Overall, I was concerned about the number of statistical tests and the number of factors considered in this relatively limited dataset. Consider eliminating tests that are not clearly related to the hypotheses, or identifying those tests that were planned vs. exploratory. I strongly recommend consulting a statistician. Also consider including a rationale for testing difference stimulation modes and apical-vs-basal electrodes, with hypotheses and/or additional considerations.

Detailed comments

page 2, bottom: Based on this abstract, a reader would probably assume that rate pitch improved for all stimulation rates, including 400 and 800 Hz. This should be clarified.

line 32: Consider citing something that describes the controversy surrounding the maximum phase locking rate in humans. e.g., Verschooten, E., Shamma, S., Oxenham, A.J., Moore, B.C.J., Joris, P.X., Heinz, M.G., and Plack, C.J., The upper frequency limit for the use of phase locking to code temporal fine structure in humans: A compilation of viewpoints. Hear Res, 2019. 377: 109-121.

line 44: I am not sure I would characterize this as “likely”. Consider changing this to “possible”.

line 49: There is disagreement of number between “rates” and “it”.

Table 1: It is redundant to include age at implantation, age at test, and years of implant use. If you do retain that third column, “user” should be “use”.

line 103: This wording is a little unusual. Consider changing, “an alternative forced-choice” to “a forced-choice”.

line 152: At this point I wondered how well these functions characterized the data.

line 167: Estimating thresholds based on an odd number of reversals introduces bias.

line 199: Some of the text at the beginning of each section seems redundant with the methods. In any case, consider starting with the outcome of primary interest – rate pitch discrimination.

line 316: Could there be order effects? I am not convinced that these data provide insight, as opposed to reflecting measurement noise.

line 347: At this point I became increasingly concerned about statistical power.

line 426: It was not clear to me how to interpret “removing across-subject variability”.

line 432: This study about integration and rate discrimination seems important, given that an association between these measures underlies the research design. Consider talking more about this dataset.

line 468: Text in the introduction might lead a reader to assume that nothing like this has been done. Consider mentioning this study in the introduction.

line 490: This seems to assume that learning would take place under the right conditions. I am not sure that is a valid assumption.

line 499: Consider specifying that this occurred for base rates of 100 and 200 Hz.

6. PLOS authors have the option to publish the peer review history of their article (what does this mean?). If published, this will include your full peer review and any attached files.

Reviewer #1: No

Reviewer #2: No

---

## [Author Response · Author response to Decision Letter 0]

4 Jun 2020

Among the many changes the authors made in this revision, below we describe the more major changes made to improve the paper. 

The authors reworked the methods and results sections of the revisions for flow and content, while adding significant paragraphs to the introduction and discussion to provide rationale, include relevant studies and improve the introduction and explanation of the metrics of spatial tuning and temporal integration.

The authors reworked the data analysis by re-quantifying masking proportional to the peak of the masking function in Figure 4 and normalizing the metrics by the average across conditions for each subject in Figure 6.

The authors recognize the significant statistical concerns and have worked to address those by going over data analysis and statistics, removing a figure, and designating planned vs exploratory statistical analyses.

---

## [Decision Letter · Decision Letter 1]

23 Jun 2020

PONE-D-19-23873R1

Perceptual Learning of Pitch Provided by Cochlear Implant Stimulation Rate

PLOS ONE

Dear Dr. Bissmeyer

Thank you for submitting your manuscript to PLOS ONE. After careful consideration, we feel that it has merit but does not fully meet PLOS ONE’s publication criteria as it currently stands. Therefore, we invite you to submit a revised version of the manuscript that addresses the points raised during the review process.

The two reviewers have commented on your revision. After reading the reviews and looking at the manuscript, I am afraid that I have to concur with the more critical review. Please pay attention to the following reviewer suggestions and give them due consideration.

- Crucially the point raised by the reviewer about the statitical analyses needs to be fully addressed.

You must adequately address the statistical analyses presented (with may impact on the conclusions drawn within the paper), otherwise your paper cannot be accepted.

- Please also provide a clear structured response to the reviewer comments.

We look forward to receiving your revised manuscript.

Kind regards,

Ifat Yasin

Academic Editor

PLOS ONE

Reviewers' comments:

Reviewer's Responses to Questions

**Comments to the Author**

1. If the authors have adequately addressed your comments raised in a previous round of review and you feel that this manuscript is now acceptable for publication, you may indicate that here to bypass the “Comments to the Author” section, enter your conflict of interest statement in the “Confidential to Editor” section, and submit your "Accept" recommendation.

Reviewer #1: All comments have been addressed

Reviewer #2: (No Response)

2. Is the manuscript technically sound, and do the data support the conclusions?

Reviewer #1: Yes

Reviewer #2: Partly

3. Has the statistical analysis been performed appropriately and rigorously? 

Reviewer #1: Yes

Reviewer #2: No

4. Have the authors made all data underlying the findings in their manuscript fully available?

Reviewer #1: Yes

Reviewer #2: Yes

5. Is the manuscript presented in an intelligible fashion and written in standard English?

Reviewer #1: Yes

Reviewer #2: Yes

6. Review Comments to the Author

Reviewer #1: I recommend that the paper be accepted for publication - the paper has considerably improved after the revision and the authors were extremely responsive to my previous comments. The findings were of high important to the field. It is fascinating that rate discrimination is so trainable and the results provide strong evidence for using variable rates.

Just a few minor remaining issues:

Line 225: please provide rationale for using logarithmically transformed thresholds.

Legend of Figure 4: this seems to be the only place where normalization was described. A little more details are needed. Was proportaiotn of percent dynamic range calculated in linear scale or log scale?

Line 303: combine the two sentences

Line 474: should be the "shallowest slopes"

Line 482: needs a little more elaboration

Reviewer #2: My primary comment on this manuscript is the same one I had on the previous version. I am concerned that the modest n does not support the extensive statistical analysis. It is problematic to run an ANOVA with a four-way interaction with n=7. I previously suggested that the authors consult a statistician, and I continue to believe that is the best way forward. A structured response to reviewer comments would also be helpful.

7. PLOS authors have the option to publish the peer review history of their article (what does this mean?). If published, this will include your full peer review and any attached files.

Reviewer #1: Yes: Ning Zhou

Reviewer #2: No

---

## [Author Response · Author response to Decision Letter 1]

6 Nov 2020

Dear Dr. Ifat Yasin,

The authors thank you and the reviewers for the helpful comments provided for our paper throughout this revision process. In the current revision, we have addressed all comments – both specific and general – from both reviewers. Reviewer #1 had minor changes, which we addressed in the manuscript and comment on below. Reviewer #2 strongly recommended consulting a statistician. We consulted a statistician who provided two main recommendations.

The statistician’s first main recommendation was to clearly describe our data and intended statistical analysis in the methods section. The statistician commented that the strength of the collected data is that the data set contains repeated measures and was a full-factorial design, two things that were not clear in the original manuscript. The revision provides a clearer picture of the depth and fullness of the data set which supports the first level of hypothesis testing. 

The statistician’s second main recommendation was to remove third order and higher considerations from the analyses. She commented that the depth of the data, having repeated measures and of full-factorial design, provides power to consider higher order interactions, but that doing so ultimately detracted from the analysis. As this was in line with the reviewer’s concerns, we fully agreed and consequently removed third-order and higher analyses from the manuscript.

Here we provide a structured account of manuscript changes:

I. Introduction: In the previous revision, the authors added rationale for considering stimulation modes and electrode locations and for considering metrics of spatial tuning and temporal integration. In this revision, the authors made minor word changes for clarity.

II. Methods: In this revision, we added an analysis subsection to the methods section (section II. C.) that delineates the experimental design and planned analyses.

III. Results: In this revision, the statistician recommended that we remove the 3rd and higher order interactions from the statistical analysis for rate discrimination. The analysis of variance was conducted with second-order interactions; the categorical statements of significance have not changed from the prior submission.

IV. Discussion: In the previous revision, we added multiple paragraphs to the Discussion introducing the discussion, addressing limitations to the study, and discussing trends seen among metrics. In the current revision, we change some minor wording issues and added a few sentences at lines 488-494 addressing a trend among the measures.

Below the author’s address specific reviewer comments:

Reviewer #1: I recommend that the paper be accepted for publication - the paper has considerably improved after the revision and the authors were extremely responsive to my previous comments. The findings were of high important to the field. It is fascinating that rate discrimination is so trainable and the results provide strong evidence for using variable rates.

Author response: Thank you very much!

Just a few minor remaining issues:

Line 225: please provide rationale for using logarithmically transformed thresholds.

Author response: Thank you, we have provided a reference (Micheyl et al. 2006) to the rationale for logarithmically transformed thresholds at lines 220-222 beginning with “All statistics were calculated…”.

Legend of Figure 4: this seems to be the only place where normalization was described. A little more details are needed. Was proportion of percent dynamic range calculated in linear scale or log scale?

Author response: We now describe the normalization in the results at line 305-307 beginning with “Thresholds were normalized in linear scale by dividing the forward masking…”. The normalization, or proportion of %DR was calculated in linear scale.

Line 303: combine the two sentences

Author response: Thank you, we have done so at line 316-318.

Line 474: should be the "shallowest slopes"

Author response: Thank you for this comment, we have changed this at lines 479 and 489.

Line 482: needs a little more elaboration

Author response: Thank you for this comment. We have added two sentences to the discussion section to elaborate on the possible explanation for the surprising trend that was seen. These sentences begin with “This trend is in agreement with Zhou and colleagues (2019) with the sites exhibiting the shallowest forward-masked thresholds slopes…” at lines 488-494.

Reviewer #2: My primary comment on this manuscript is the same one I had on the previous version. I am concerned that the modest n does not support the extensive statistical analysis. It is problematic to run an ANOVA with a four-way interaction with n=7. I previously suggested that the authors consult a statistician, and I continue to believe that is the best way forward. A structured response to reviewer comments would also be helpful.

Author response: Thank you for your comments. We have consulted with a statistician who indicated that her primary recommendation was to clearly describe our data and intended analysis in the methods section. The statistician commented that the strength of the collected data is that the data set contains repeated measures and was of full-factorial design, two things that were not clear in the original manuscript. The revision provides a clearer picture of the depth and fullness of the data set which supports the first level of hypothesis testing. The statistician seconded the concern with third and higher-order interactions, which we have removed.

---

## [Editor Report · Decision Letter 2]

11 Nov 2020

Perceptual Learning of Pitch Provided by Cochlear Implant Stimulation Rate

PONE-D-19-23873R2

Dear Dr. Bissmeyer,

We’re pleased to inform you that your manuscript has been judged scientifically suitable for publication and will be formally accepted for publication once it meets all outstanding technical requirements.

Kind regards,

Ifat Yasin

Academic Editor

PLOS ONE
---

## [Editor Report · Acceptance letter]

24 Nov 2020

PONE-D-19-23873R2 

Perceptual Learning of Pitch Provided by Cochlear Implant Stimulation Rate 

Dear Dr. Bissmeyer:

I'm pleased to inform you that your manuscript has been deemed suitable for publication in PLOS ONE. Congratulations! Your manuscript is now with our production department. 

Kind regards, 

on behalf of

Dr. Ifat Yasin 

Academic Editor

PLOS ONE